# Identification of an Individualized Prognostic Biomarker for Serous Ovarian Cancer: A Qualitative Model

**DOI:** 10.3390/diagnostics12123128

**Published:** 2022-12-12

**Authors:** Fengyuan Luo, Na Li, Qi Zhang, Liyuan Ma, Xinqiao Li, Tao Hu, Haijian Zhong, Hongdong Li, Guini Hong

**Affiliations:** 1School of Medical Information Engineering, Gannan Medical University, Ganzhou 341000, China; 2Affiliated Hospital of Jiangxi, University of Chinese Medicine, Nanchang 330006, China

**Keywords:** serous ovarian cancer, relative expression orderings, prognostic biomarker

## Abstract

Serous ovarian cancer is the most common type of ovarian epithelial cancer and usually has a poor prognosis. The objective of this study was to construct an individualized prognostic model for predicting overall survival in serous ovarian cancer. Based on the relative expression orderings (Ea > Eb/Ea ≤ Eb) of gene pairs closely associated with serous ovarian prognosis, we tried constructing a potential individualized qualitative biomarker by the greedy algorithm and evaluated the performance in independent validation datasets. We constructed a prognostic biomarker consisting of 20 gene pairs (SOV-P20). The overall survival between high- and low-risk groups stratified by SOV-P20 was statistically significantly different in the training and independent validation datasets from other platforms (*p* < 0.05, Wilcoxon test). The average area under the curve (AUC) values of the training and three validation datasets were 0.756, 0.590, 0.630, and 0.680, respectively. The distribution of most immune cells between high- and low-risk groups was quite different (*p* < 0.001, Wilcoxon test). The low-risk patients tended to show significantly better tumor response to chemotherapy than the high-risk patients (*p* < 0.05, Fisher’s exact test). SOV-P20 achieved the highest mean index of concordance (C-index) (0.624) compared with the other seven existing prognostic signatures (ranging from 0.511 to 0.619). SOV-P20 is a promising prognostic biomarker for serous ovarian cancer, which will be applicable for clinical predictive risk assessment.

## 1. Introduction

Ovarian cancer is one of the three major malignant tumors of the female reproductive system, and its incidence ranks third among gynecological malignancies [1,2]. However, because the ovaries are deep in the pelvis, the onset is insidious, and the early symptoms are not apparent [3]. As the most common ovarian epithelial tumor subtype, serous ovarian cancer accounts for about 40–50% of all malignant ovarian tumors, and its survival rate within five years is only about 30% [4]. Clinically, physicians treat serous ovarian cancer patients with the same clinical and pathological stages in the same way. However, due to the heterogeneity of the tumor, patients usually have different survival outcomes. Studies have shown clinical prognostic factors such as the stage of The Federation of Gynecology and Obstetrics (FIGO), the size of residual after debulking surgery, BRCA1/2 mutations, and the degree of lymphocyte infiltration could cause gene expression changes in serous ovarian cancer [5,6]. These influencing factors are related to the high recurrence rate and poor prognosis of serous ovarian cancer through the activation of oncogenes and the inactivation of tumor suppressor genes [7]. Therefore, there is an urgent need to explore gene expression biomarkers that can predict the prognostic effect of serous ovarian cancer.

In recent years, many studies have used gene expression information to develop biomarkers for serous ovarian cancer [8]. The level of patient risk was determined by a predetermined threshold of the risk score in the models [9,10,11]. However, such a risk-score-based prognostic model has some limitations. Firstly, gene expression levels are sensitive to systematic bias in high-throughput data [12,13]. The risk thresholds generated by the training dataset cannot be directly applied to independent datasets. Therefore, the training and validation data need to be pre-normalized to make the trained thresholds applicable to new data [14]. This means that the risk of samples in the training data can significantly influence the risk prediction outcome of an unknown sample. In addition, data normalization requires prior sample collection, which may be clinically impractical. However, without data normalization, the risk scores of independent samples are not comparable [15,16]. Therefore, finding stable prognostic risk biomarkers with clinical translational value in serous ovarian cancer samples has become an urgent issue.

Several studies have developed disease-related biomarkers based on the within-sample relative expression orderings (REOs) of genes [17,18,19]. Studies have claimed that REOs are insensitive to systematic biases [20,21]. They are also robust against interindividual biological variation of gene expression levels. Moreover, REOs use qualitative information about gene expression. They relate only to relative changes in gene expression within individual samples.Thus, they are independent of other samples, indicating that REO-based biomarkers can be naturally applied to individual samples [22]. Thus, our objective in this study was to develop an individualized prognostic model for serous ovarian patients based on the within-sample REOs of genes.

## 2. Materials and Methods

### 2.1. Study Population

Gene expression profiles from eight independent array datasets containing 1493 samples were collected from Gene Expression Omnibus (GEO, http://www.ncbi.nlm.nih.gov/geo/, accessed on 11 October 2020) and the UC Santa Cruz Xena platform (UCSC Xena, https://xenabrowser.net/datapages/, accessed on 1 November 2020). The division of data into training, test, and validation sets is shown in Figure 1. Training datasets included 561 samples divided into low-grade (G1, *n* = 6), high-grade (G2–G4, *n* = 544), borderline (*n* = 9), and ungraded (*n* = 2). Clinical information, including age, grade, and stage, was available for most datasets. The detailed clinical information for each dataset can be found in Table 1. Of all the data used, chemotherapy information was only available for the 466 samples from TCGA. Detailed tumor response information for TCGA samples was provided in Appendix A.

The RMA algorithm pre-processes the raw microarraydata downloaded from GEO. The annotation files in each platform were used to map probe IDs to gene IDs. Probes that were not mapped to genes were deleted. For different probes mapped to the same gene, the average was used as the final expression value of the gene Normalized gene expression profiling data and clinical data from 630 TCGA ovarian cancer patients were downloaded from the UCSC Xena platform. The data were pre-processed in the following steps before constructing a prognostic biomarker: (1) remove normal samples; (2) remove tumor samples without clinical information; (3) remove samples with overall survival of 0 days; and (4) remove low-expression genes (more than half of the samples with missing or 0 gene expression).

### 2.2. Identification of Candidate Prognosis-Related Gene Pairs

Univariate Cox regression analysis was used to evaluate the predictive value of genes and gene pairs. When Cox regression analysis was applied to single genes, a *p*-value less than 0.05 was considered significantly associated with patients’ overall survival. Pairing every two significant prognosis-related genes yields candidate prognosis-related gene pairs. Then, the REO matrix *X* of the candidate prognosis-related gene pairs was constructed. *X* is a 0–1 matrix, wherein *x_ij_* = 1 indicates that the REO of the gene pair *i* (*G_a_, G_b_*) in the serous ovarian cancer sample *j* is *G_a_* > *G_b_,* and *x_ij_*= 0 indicates *G_a_* ≤ *G_b_*. When Cox regression analysis was applied to gene pairs, an adjusted *p*-value less than 0.05 after the Benjamini–Hochberg correction was considered significant [23].

### 2.3. Identification of The Prognostic Biomarker

Given the concordance index (C-index) values for all gene pairs in the significant prognosis-related gene pairs, we selected the prognostic biomarker comprising top-ranked gene pairs in a greedy manner. First, the first *N* gene pairs with the highest C-index were selected. Then, for each top *N* gene pair, the remaining significant prognosis-related gene pairs were added to form the combination if the added gene pairs could increase the C-index values of the combination. For each of the *N* combinations, the test set GSE13876 was used to find the combination significantly correlated with the prognosis of serous ovarian cancer with a higher C-index value as the final prognostic-related biomarker gene pairs.

### 2.4. Performance Evaluation of The Prognostic Biomarker

A sample was classified into the high- or low-risk group based on the majority voting of the gene pairs in the prognostic biomarker by their REOs. Gene pairs not detected in cross-platform validation data were removed, and the remaining gene pairs were used to determine the risk labels.

Multivariate Cox risk regression analyses wereconducted to evaluate the joint prognostic significance of the developed prognostic biomarker and clinical factors, including age, stage, and grade. As clinical information was unavailable in all datasets (see Materials and methods, Table 1), we only included the existing clinical elements in each dataset. The overall survival was calculated using the Kaplan–Meier, and the log-rank testwas used to compare the two survival curves. Forest plots were used to display the results of multiple Cox regression analyses.

### 2.5. Functional Enrichment Analysis

Gene annotation and pathway analysis were performed in the database of Metascape (http://www.metascape.org/) to search for functions associated with genes in the prognostic biomarker. We considered the following ontology sources in Metascape: KEGG Pathways, GO Biological Processes, Reactome Gene Sets, Canonical Pathways, CORUM, TRRUST, DisGeNET, PaGenBase, Transcription Factor Targets, Wiki Pathways, and COVID. To further understand the biological function of the differentially expressed genes between the high- and low-risk groups predicted by the prognostic biomarker, functional enrichment analysis was carried out based on the Kyoto Encyclopedia of Genes and Genomes (KEGG). The relevance was considered significant if the Wilcoxon *p*-value was less than 0.05.

### 2.6. Immune Infiltration Analysis

TIMER2.0 (http://timer.cistrome.org/, accessed on 10 September 2021) was used to quantify the proportions of immune cell infiltration, and the differences between the high- and low-risk groups were compared. The content of six immune cells (B cells, CD4+ T cells, CD8+ T cells, neutrophils, macrophages, and dendritic cells) was estimated, and the differences were compared using the Wilcoxon test method.

### 2.7. Drug SensitivityAnalysis

The chemotherapeutic response for each serous ovarian cancer sample in the training sets was predicted according to the *Genomics of Drug Sensitivity in Cancer* (GDSC, https://www.cancerrxgene.org/, accessed on 24 November 2022) database. The chemotherapy sensitivity of each tumor sample was evaluated by the half-maximal inhibitory concentration (IC50) value, which was obtained using regression analysis, and the prediction accuracy was measured by 10-fold cross-validation. The differences in IC50 values between high- and low-risk groups were compared using the Wilcoxon test method.

### 2.8. Performance Comparison with Other Prognostic Biomarkers

Seven published prognostic models based on gene expression data were selected [24,25,26,27,28,29]. Since most of the models are constructed based on TCGA-OV sequencing data, the RNA sequencing data of TCGA-OV in UCSC Xena are collated to ensure comparativeness. According to the median risk score of each model, the samples were also split into high- and low-risk groups, and the overall survivals were compared.

### 2.9. Statistical Analysis

Statistical analyses were performed using R (version 4.2.0). We used the *survival* R package for univariate and multivariate risk regression analysis. The *timeROC* and *survival* packages were used to evaluate the receiver operating characteristic curve (ROC curve) and calculate area under curve (AUC) values. The *survminer* and *survival* R packages were used to show the K-M curves and Forest plots of grouped samples. The *ggplot2* and *clusterProfiler* packages were used to enhance the appearance of the enrichment plot, and *ggplot2* was also used to draw a violin boxplot diagram. The *oncoPredict* was used to determine the IC50 for evaluating drug sensitivity. All parameters are default, including “*combat*”, to eliminate the batch effect and the average of repeated gene expression. The *rms* R package was used for calculating the C-index.

## 3. Results

### 3.1. Identification of the Prognosis-Related Biomarker

To detect the prognosis-related gene pairs, we first performed the Cox regression to find genes that are significant predictors of survival. Only genes common to all platforms in the training set were considered. There were 6934 genes in common, of which 82,550, and 796 were significantly correlated with overall survival in GSE18520, GSE19829, and TCGA-OV data, respectively (*p* < 0.05, Wilcoxon test, Appendix A). Among them, 1347 were significant in at least one of the three datasets, and 486 genes had consistent directions of hazard coefficients (positive or negative). These 486 genes were selected for subsequent analysis. After pairing the 486 genes, we obtained 117,855 prognosis-related gene pairs, and 11,569 were significantly associated with overall survival (*p* < 0.05, Wilcoxon, BH adjusted for multiple testing).

Then, the prognosis-related biomarker was developed by applying the greedy method to the top gene pairs with the largest C-index values (see Materials and Methods). Here, we used the top 10 gene pairs (C-index ranging from 0.68 to 0.72) to form the final combinations of selected gene pairs. The test dataset was used to obtain the best combination, and univariate Cox regression analysis was performed for each of the ten combinations. The combination significantly correlated with overall survival with the largest C-index value (*p* = 0.00046, Wilcoxon test; C-index = 0.69) was selected as the final prognostic biomarker, which contains 20 gene pairs (Appendix A), and referred to as SOV-P20.

### 3.2. Prediction of Overall Survival by SOV-P20

In the training set, 298 and 452 samples were categorized into high- and low-risk groups, respectively. The high-risk group had a significantly decreased overall survival compared to the low-risk group (Figure 2A, *p* < 0.0001, HR = 0.23,95% CI: 0.18–0.29). In the three validation datasets, 292 and 136, 44 and 145, and 129 and 150 samples were divided into high- and low-risk groups, respectively. Kaplan–Meier survival analyses showed that in all datasets, the high-risk group had a significantly lower overall survival rate. In contrast, the low-risk patients usually showed a longer survival time, with a significant difference between the two groups (*p* = 0.0077, HR = 0.60, 95% CI: 0.41–0.88, Figure 2B; *p* = 0.028, HR = 0.54, 95% CI: 0.31–0.94, Figure 2C; *p* = 0.0062, HR = 0.59, 95% CI: 0.41–0.87, Figure 2D).

Since the overall survival time of patients exceeded five years, the AUCs of SOV-P20 for 3, 5, and 7 years were calculated. The mean AUC was 0.756 in the training set, while in the three validation datasets, the mean AUCs were 0.590, 0.630, and 0.680 (Figure 2E–H), respectively, indicating a potential predictive value of the SOV-P20 biomarker.

Multivariate Cox analyses were conducted to confirm whether the SOV-P20 was an independent prognostic factor. The results showed that age and SOV-P20 were significantly associated with overall survival (*p* < 0.05, log-rank test, Figure 2I–L), and SOV-P20 was an independent risk factor for patients with serous ovarian cancer.

### 3.3. Functional Enrichment Analysis

Functional enrichment analysis was performed on the 37 genes comprising SOV-P20. As shown in Figure 3, eight GO terms related to cytokine production were significantly enriched (Figure 3A). Cytokine refers to small molecule polypeptides secreted primarily by immune cells that regulate cell function [30]. Cytokines and cytokine receptor processes are mainly related to regulating the body’s immune response, hematopoietic function, and inflammatory response [31]. They can inhibit tumorigenesis and progression and have been shown to be effective in cancer treatment [32]. Three genes are involved in the virus’s entry into the host cell biology pathway. Pathway analysis showed that five genes were associated with tumor recurrence (Figure 3B), confirming that SOV-P20 plays a role incancer onset and development.

Further enrichment analysis for differentially expressed genes between the high- and low-risk groups identified three significant KEGG pathways (BH adjusted *p* < 0.05), including PI3K-Akt signaling pathway, proteoglycan pathway, and AGE-RAGE signaling pathway in cancer (Figure 3C). These pathways have been reported to play an essential role in the progression and prognosis of serous ovarian cancer [33,34,35].

### 3.4. Immune Infiltration Analysis

The distribution of tumor-infiltrating immune cells is an essentialindicator of immune invasion and prognosis in patients [36]. Results from TIMER showed significant differences in the infiltration levels of immune cells between the high- and low-risk groups (*p* < 0.05, Wilcoxon test, Figure 3D), and the low-risk group had higher infiltration levels.

### 3.5. Tumor Response to Drug Treatment

For all training samples, only TCGA data provide information about an individual patient’s response to chemotherapy drugs (Appendix A). The RECIST was used to evaluate tumor response to chemotherapy drugs, categorized as complete response (CR), partial response (PR), stable disease (SD), or progressive disease (PD). Only 338 patients with RECIST measurements were analyzed. Among them, 143 were high-risk, and 195 were low-risk. They were further classified as chemo-sensitive (CR and PR) or chemo-resistant (SD and PD). Of the 143 high-risk patients, 107 were chemo-sensitive, and 36 were chemo-resistant. In contrast, of the 195 low-risk patients, 171 were chemo-sensitive, and 24 were chemo-resistant, significantly different from the high-risk groups (Fisher’s exact test, *p* = 2.50 × 10^−3^). In addition, compared to high-risk patients, low-risk cases tended to be chemo-sensitive (odds ratio = 2.397) and were less likely to be chemo-resistant (odds ratio = 0.417).

Similar analyses were performed for 310 TCGA patients who received platinum-based chemotherapy. There was a significant difference in sensitivity and resistance to platinum-based chemotherapy between the high- and low-risk groups (Fisher’s exact, *p* = 6.59 × 10^−4^), as the number of chemo-sensitive and chemo-resistant patients were 100 and 33 for high-risk groups, respectively, while in low-risk groups, the numbers were 157 and 20, respectively. Low-risk cases were more likely to be chemo-sensitive to platinum (odds ratio = 2.591) and less inclined to develop chemoresistance to platinum (odds ratio = 0.386) relative to the high-risk group.

Since only TCGA samples had drug-response information, we used the GDSC database to predict the response to commonly used chemotherapeutic drugs for each training sample. As shown in Figure 3E, patients in the low-risk groups were more sensitive to Cisplatin, Gemcitabine, Oxaliplatin, Sorafenib, Tamoxifen, and Topotecan. In contrast, patients in the high-risk groups were more sensitive to 5-Fluorouracil, and there were significant differences between the high- and low-risk groups (*p* < 0.05, Wilcox test).

The above results revealed a significant difference in tumor response to chemotherapy between the high- and low-risk groups predicted by SOV-P20.

### 3.6. Comparison with Other Models

To further validate the predictive value of SOV-P20, we compared its performance with seven different prognostic models previously reported using TCGA-OV RNA sequencing data. Survival was significantly different between high- and low-risk groups in TCGA-OV samples assessed by the seven models, and the high-risk group had a poor prognosis (Figure 4A, all *p* < 0.05). The C-index of the above seven prognostic models was smaller than the C-index of SOV-P20 (Figure 4A), indicating that the overall predictive performance of the prognostic model in this paper was better than that of the other seven models in other studies.

The performance comparison of SOV-P20 with the seven predictive models was also performed using the validation set collected in this study. In the seven models, only the genes detected in each dataset were analyzed. Results showed that, except for the 8-gene model by Zhang et al. [37] with a significant Kaplan–Meier curve in the validation set 2 (*p* = 0.0083), all Kaplan–Meier curves of the remaining prognostic models in each validation dataset showed no statistically significant difference between high- and low-risk groups (Figure 4B). SOV-P20 achieved the highest mean C-index (0.624) compared with the other six existing prognostic signatures (ranging from 0.511 to 0.619, Figure 4C). The above results further suggested that the prognostic biomarker developed in this study had a robust predictive ability.

## 4. Discussion

### 4.1. Main Findings

In this study, we constructed the gene-pair-based biomarker, SOV-P20, to predict the prognosis of serous ovarian cancer and further validated the predictive value in independent datasets. Our results suggest that SOV-P20 has strong robustness and stable predictive performance in datasets from different platforms. Many genes involved in SOV-P20 have been reported as prognostic biomarkers or potential therapeutic targets for serous ovarian cancer. The literature supporting the association between genes in SOV-P20 and serous ovarian cancer is shown in Appendix A. Genes in SOV-P20 were significantly enriched in the biological pathways related to regulating cytokine production, virus entry into host cells, and tumor recurrence. The high- and low-risk groups predicted by SOV-P20 showed a significant difference in overall survival time and tumor-infiltrating immune cells. Differentially expressed genes between the high- and low-risk groups were significantly enriched in the PI3K-Akt signaling pathway, the proteoglycan pathway, and the AGE-RAGE signaling pathway in cancer. Compared with the published prognostic models, SOV-P20 stood out in terms of C-index, demonstrating its good predictive ability for ovarian cancer prognosis.

### 4.2. Strengths and Limitations

Many studies have developed genetic biomarkers for stratifying patient survival on different datasets. However, these biomarkers have not been applied in the clinical setting due to various limitations. There are many reasons. Firstly, most researchers agree that developing a robust prognostic biomarker requires a large sample size. Most studies have a relatively small sample size, which may suffer from model overfitting problems. As shown in this study, by applying Cox regression to the three datasets in the training set, the prognosis-related genes were detected and intersected, and few intersection genes were found (Appendix A). Therefore, when developing the biomarker, we combined 561 samples into the training set to meet the requirement of large sample size while trying to include data from different platforms. Second, most current predictive biomarkers for patient stratification are based on risk scores, which are sensitive to system deviations and poorly robust. To address this issue, we developed the predictive model based on the REOs within each sample. As a qualitative model, it is highly robust and insensitive to systematic bias and laboratory batch effects. The prognostic value of the developed biomarker was confirmed in the test set of 415 samples and the validation set of 517 samples from different platforms.

However, there are some limitations to this study. First, the method of identifying disease biomarkers based on within-sample REOs is a qualitative metric. Considering only REOs of genes may lose some subtle quantitative information about gene expression. Second, the imperfect clinical characteristics of the dataset may reduce the accuracy of multivariate survival analysis and comprehensive prediction model when performing multivariate Cox regression analysis. Third, genes in the prognostic biomarker were not subjected to a wet biological experiment, which deserves continued study in detail in future work.

### 4.3. Interpretation of Findings

Many studies have used gene expression information to develop predictive biomarkers or diagnostic models for serous ovarian cancer (Appendix A). We divided these studies into two main categories according to the number of feature genes.

In the first category, researchers assessed the impact of individual genetic biomarkers on the prognosis of ovarian cancer. For example, Buttarelliet al. identified the long-chain noncoding RNA-MEG3 gene as a tumor suppressor for serous ovarian cancer through in vitro and in vivo experiments, demonstrating that its high expression predicts better progression-free survival and overall survival [8]. A study by Zhang et al. showed that miR-212-3p might suppress serous ovarian cancer by directly targeting MAP3K3 [38]. However, such single-gene-based studies do not consider the effects of gene-to-gene interactions and might not accurately predict the prognosis of patients with serous ovarian cancer.

The second category evaluates the predictive value of a set of genes based on a specific biological function. For example, Yang et al. constructed a risk scoring model containing seven genes to predict the prognosis of serous ovarian cancer based on stromal and immune infiltrates in the immune tumor microenvironment [39]. Wang et al. constructed an 8-gene predictive risk assessment model for serous ovarian cancer from energy metabolism-related genes [28]. Zheng et al. built a prognostic risk scoring model for serous ovarian cancer using 11 lipid metabolism-related genes [11]. However, such biological function-based predictive biomarkers may ignore the heterogeneity of individual patients and the complexity of influencing factors and are prone to overfitting.

We found more than 20 published prognostic models for serous ovarian cancer (see Appendix A). However, many models’ genes could appear undetected when applied to datasets from different platforms. In this study, we did not compare the single-gene-based biomarkers and only selected those models whose comprising genes were detected across platforms for comparison. Finally, only seven models were included in the comparison. We found that the Kaplan–Meier curve results for six models were not statistically significant enough to predict the prognosis in independent datasets other than their own. According to our study, when the median risk score of the training set was taken as the threshold, the test set could not make prognostic predictions. In contrast, SOV-P20 was developed from samples from different platforms, and it can still predict the prognosis on the principle of half voting when the platform does not detect all genes. Actually, SOV-P20 can be applied to any data set where no thresholds are needed. Moreover, it is REO-based and thus robust to noise in the data and offers a natural way to overcome the heterogeneity across datasets, especially across different platforms. Therefore, SOV-P20 can achieve the goal of personalized precision medicine.

Most serous ovarian cancer samples analyzed in this study were high-grade. Currently, the standard first-line care for high-grade serous ovarian cancer remains surgical resection and platinum-based chemotherapy. We analyzed the tumor response to platinum-based chemotherapy. The results showed a significant difference in sensitivity to platinum-based chemotherapy between the high- and low-risk groups. Notably, resistance to platinum-based chemotherapy was present in both high- and low-risk groups, consistent with the assumption that a small proportion of ovarian cancer cells are chemo-resistant from the beginning [40]. Such inherent resistance in ovarian cancer may occur due to reduced immunosurveillance and drug-resistant cells that belong to cancer stem cells (CSCs) [7,40]. CSCs from the ovaries have a particular genetic signature that can engineer the original tumor mass, develop drug resistance, and promote recurrence. In addition to this possibility, platinum resistance may also emerge due to reduced intracellular drug accumulation, intracellular inactivation of the agent, increased DNA repair, or impaired apoptotic signaling pathways [41]. Therefore, the prediction of a tumor to platinum chemotherapy in ovarian cancer will provide further research directions in the future.

The study only used the retrospective data generated on a genome-wide scale. In the future, we will use real-time PCR (RT-PCR) techniques to validate SOV-P20 on a separate ovarian cancer patient cohort prospectively. We will recruit patients with ovarian cancer in the First Affiliated Hospital of Gannan Medical University. After obtaining informed consent from each patient according to the protocol approved by the Ethics Committee of the hospital, the Department of Obstetrics and Gynecology will admit patients diagnosed with ovarian cancer between September 2021 and October 2023. Two hundred patients will be enrolled. Patient inclusion criteria include patients with a diagnosis of serous or endometrioid tumor. Specific guidelines are as follows: age 18 years or older; histologically confirmed epithelial ovarian, fallopian tube, or primary peritoneal cancer; FIGO stage I–IV; and planned to receive platinum-based combination chemotherapy. Patients will be followed up until 31 October 2028. After obtaining tissue samples, the RT-PCR techniques will determine the expression of individual mRNAs of SOV-P20 in patients to validate the predictive efficacy of SOV-P20.

## 5. Conclusions

Biomarker discovery based on within-sample REOs is a promising approach that can address the critical limitations of the risk-scoring-based model. The developed ovarian cancer prognostic model can easily predict patients with different survival prognoses. It is worth testing in prospective clinical trials to achieve its individualized management clinical effect in patients with serous ovarian cancer.

## Figures and Tables

**Figure 1 diagnostics-12-03128-f001:**
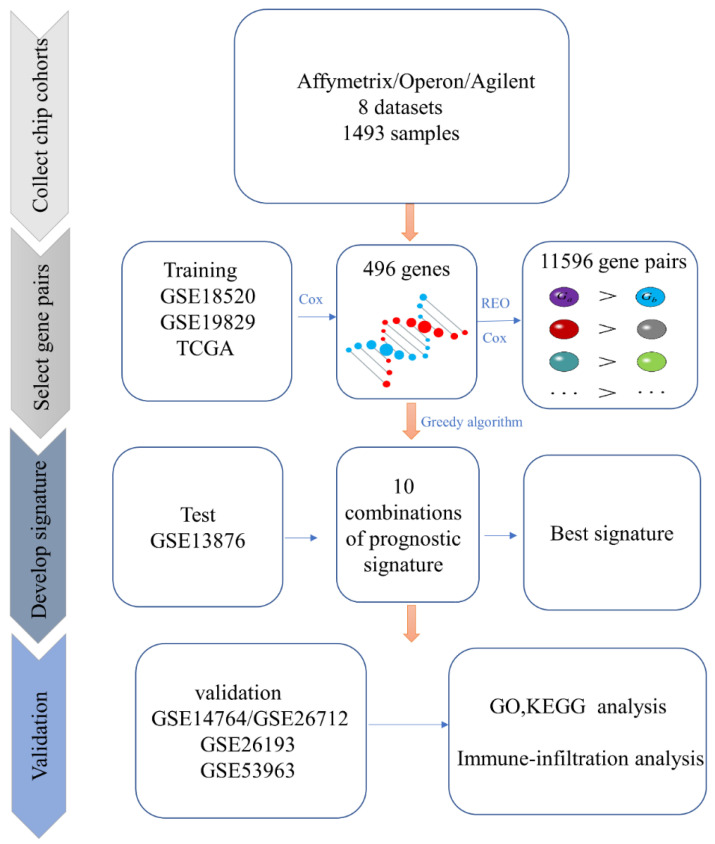
Flow chart of the study.

**Figure 2 diagnostics-12-03128-f002:**
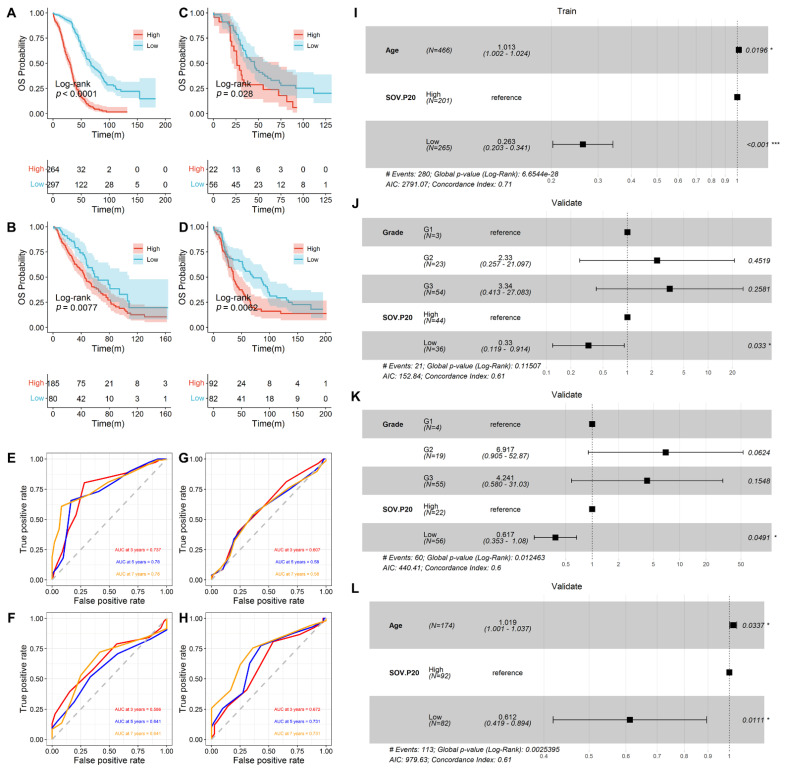
SOV-P20 predicts the high- and low-risk stratification of overall survival. (**A**–**D**) Kaplan–Meier overall survival curves of serous ovarian cancer patients in the training and three validation datasets; (**E**–**H**) the time-dependent ROC curves of SOV-P20 predict the 3-, 5-, and 7-year overall survival in the training and three validation datasets; (**I**–**L**) multivariate Cox regression analysis forest plots in the training and three validation datasets, * *p* < 0.05; *** *p* < 0.001.

**Figure 3 diagnostics-12-03128-f003:**
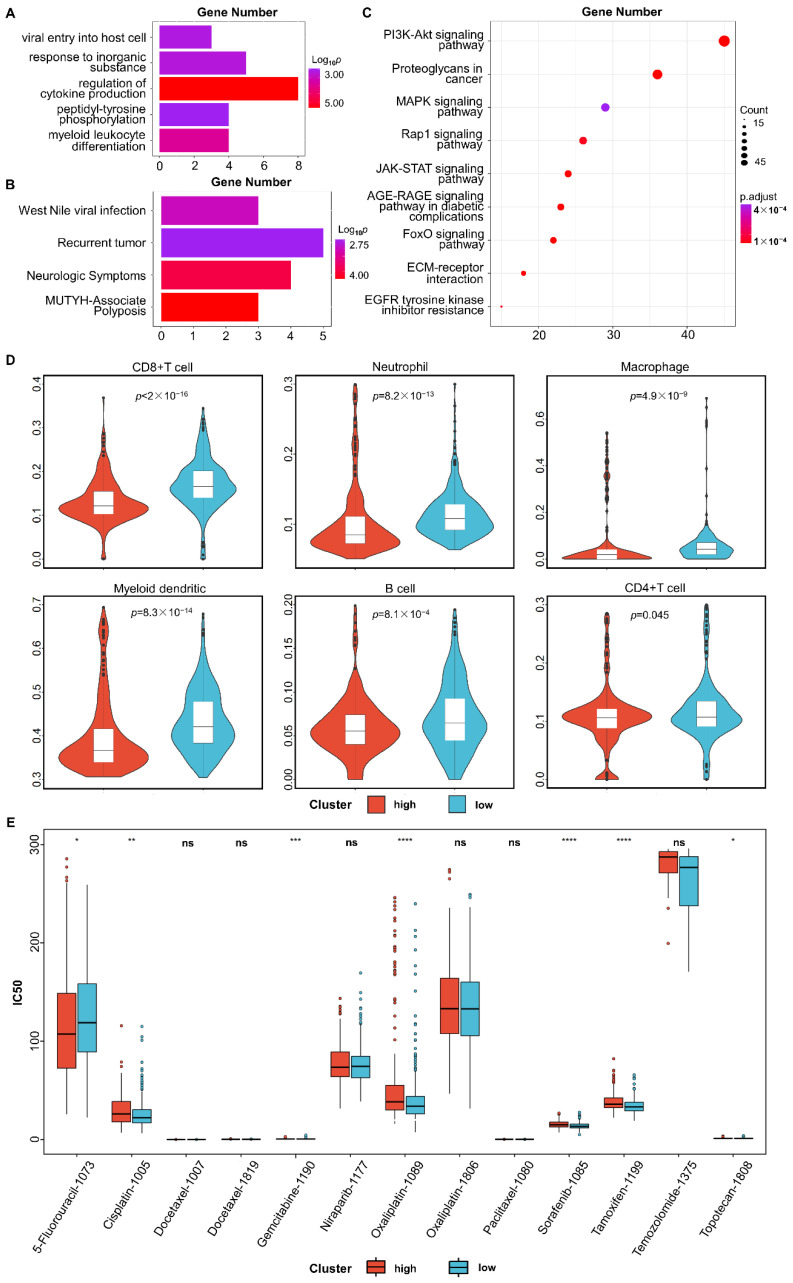
Functional enrichment and immune infiltration analysis. (**A**) Significant GO biological processes enriched by SOV-P20. (**B**) Significant DisGeNet pathways enriched by SOV-P20. (**C**) Significant KEGG pathways enriched by differentially expressed genes between the high- and low-risk groups divided by SOV-P20. (**D**) Vioplot of the difference in the infiltration scores of CD8+ T cells, neutrophils, macrophages, myeloid dendritic, B cells, and CD4+ T cells. (**E**) The difference in drug response between the high- and low-risk groups divided by SOV-P20.The legend is as following: ns, non-significant; * *p* < 0.05; ** *p* < 0.01; *** *p* < 0.001; **** *p* < 0.001.

**Figure 4 diagnostics-12-03128-f004:**
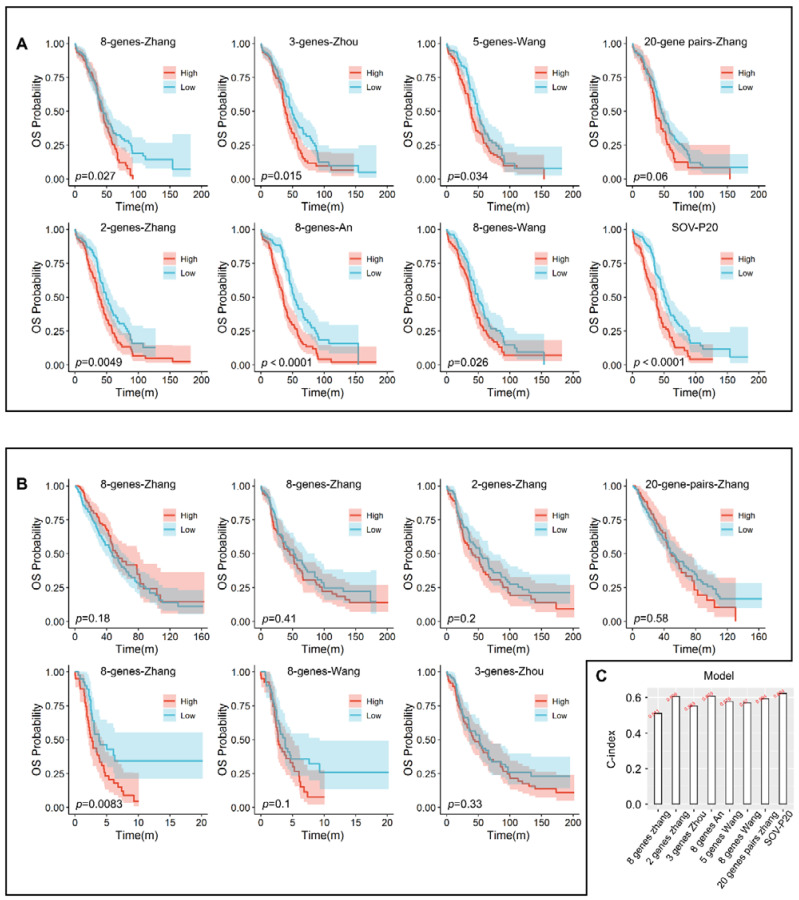
(**A**) Kaplan–Meier plots of the seven published models and SOV-P20 in TCGA-OV RNA sequencing data; (**B**) Kaplan–Meier plot of the seven published models in validation set; (**C**) bar plot of C-index values for the seven published models and SOV-P20.

**Table 1 diagnostics-12-03128-t001:** Clinical characteristics of serous ovarian cancer samples used in this study.

		Training		Test	Validation			
					Dataset 1		Dataset 2	Dataset 3
GEO accession	GSE18520	GSE19829	TCGA	GSE13876	GSE14764	GSE26712	GSE26193	GSE53963
Microarray platform	GPL570	GPL8300	GPL96	GPL7759	GPL96	GPL96	GPL570	GPL6480
Sample No.	53	42	466	415	80	185	78	174
Stage								
I	-	0	15	-	8	-	12	0
II	-	1	28	-	1	-	20	8
III	-	35	369	-	69	-	53	125
IV	-	6	51	-	2	-	14	41
Late	53 (III–IV)	-	-	415 (III–IV)	-	-	-	
Unstaged	-	-	3	-	-	-	-	-
Age, median	-	58.3	59.9	57.9	-	-	-	63
(range), y		(39–80)	(26–89)	(21–84)	-		-	(24–89)
Grade								
G1		1	5	-	3	-	7	0
G2		9	60	-	23	-	33	4
G3	-	32	389	-	54	-	67	90
G4	-	0	1	-	-	-	-	80
High-grade (G2/3/4)	53	-	-	-	-	-	-	-
Borderline	-	-	9	-	-	-	-	-
Ungraded	-	-	2	-	-	-	-	-
Survival, median (range), m	40.4 (5–150)	38.5 (1–68)	59.5 (0.27–182.70)	45.6 (1–234)	35 (7–73)	38.3 (0.72–163.80)	60.2 (0.1–133.71)	55.7 (0.3–201.61)

## Data Availability

Data were derived from the following resources available in GEO (http://www.ncbi.nlm.nih.gov/geo/, accessed on 11 October 2020) and the UC Santa Cruz Xena platform (UCSC Xena, https://xenabrowser.net/datapages/, accessed on 1 November 2020).

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
