# Peer review of "Identification of an Individualized Prognostic Biomarker for Serous Ovarian Cancer: A Qualitative Model"

_diagnostics, 2022, doi:10.3390/diagnostics12123128_

Round 1
Reviewer 1 Report
I found the article extremely interesting and well-targeted for the journal. I would like to recall 2 points:
1) A RT-PCR or IHC assessment of a new cohort can add important information to the study.
2) The role of platinum resistance should be at least discussed. And not showing information about platinum resistance limit the study.
Author Response
Dear Editor and reviewer,
We are sincerely grateful for your editorial efforts and suggestions for our manuscript and the anonymous reviewer for his/her insightful comments on our manuscript, which are very helpful in strengthening this manuscript.
Following are our point-by-point responses to the reviewer’s comments, and the reviewer’s critiques have been accommodated fully in various parts of the revised manuscript (marked up using the “Track Changes” function in MS Word, in the reply shown in BLUE color).
We hope that you will now find the revised version of our paper suitable for publication in Diagnostics.
Thank you very much.
Sincerely yours,
Guini Hong
Reviewer 1
I found the article extremely interesting and well-targeted for the journal. I would like to recall 2 points:
1) A RT-PCR or IHC assessment of a new cohort can add important information to the study.
Reply 1: Thank the reviewer for this insightful comment. We agreed that an RT-PCR or IHC assessment of a new cohort could significantly improve the manuscript. Based on this helpful comment, we carefully searched the RT-PCR or IHC data for an ovarian cancer new cohort stored in the public databases, including but not limited to GEO and ArrayExpress. However, we could not find public IHC data for ovarian cancer. For RT-PCR data, we did find four datasets with the GEO accession number GSE102180, GSE30009, GSE29702, and GSE30034. Unfortunately, these studies used the PCR assay to determine the expression of only some preselected interesting genes (for the four datasets, the number of assayed genes was 65, 380, 384, and 380, respectively), where genes in SOV-P20 appear undetected. In fact, as an exciting future area of investigation, RT-PCT assessment will be used to validate SOV-P20 prospectively in our future study, which may take a few years to collect samples and survival data. We have discussed this issue in the revised Discussion. Please see the revised manuscript.
In the revised Discussion: "The study only used the retrospective data generated on a genome-wide scale. In the future, we will use Real-time PCR (RT-PCR) techniques to validate SOV-P20 on a separate ovarian cancer patient cohort prospectively. We will recruit patients with ovarian cancer in the First Affiliated Hospital of Gannan Medical University. After obtaining informed consent from each patient according to the protocol approved by the Ethics Committee of the hospital, the Department of Obstetrics and Gynecology will admit patients diagnosed with ovarian cancer between September 2021 and October 2023. Two hundred patients will be enrolled. Patient inclusion criteria include patients with a diagnosis of serous or endometrioid tumor. Specific guidelines are as follows: age 18 years or older; histologically confirmed epithelial ovarian, fallopian tube, or primary peritoneal cancer; FIGO stage I-IV; and planned to receive platinum-based combination chemotherapy. Patients will be followed up until October 31, 2028. After obtaining tissue samples, the RT-PCR techniques will determine the expression of individual mRNAs of SOV-P20 in patients to validate the predictive efficacy of SOV-P20.".
2) The role of platinum resistance should be at least discussed. And not showing information about platinum resistance limit the study.
Reply 2: Thank the reviewer for this critical comment, which helps us to strengthen the study. In response to this comment, we have compared the tumor response to platinum chemotherapy between the high- and low-risk patients, as described in the revised Result:
"3.5 Tumor response to drug treatment
For all training samples, only TCGA data provides information about an individual patient’s response to chemotherapy drugs (Supplementary Table S1). The RECIST was used to evaluate tumor response to chemotherapy drugs, categorized as complete response (CR), partial response (PR), stable disease (SD), or progressive disease (PD). Only 338 patients with RECIST measurements were analyzed. Among them, 143 were high-risk, and 195 were low-risk. They were further classified as chemosensitive (CR and PR) or chemoresistant (SD and PD). Of the 143 high-risk patients, 107 were chemosensitive, and 36 were chemoresistant. In contrast, of the 195 low-risk patients, 171 were chemosensitive, and 24 were chemoresistant, significantly different from the high-risk groups (Fisher’s exact test, p=2.50×10-3). In addition, compared to high-risk patients, low-risk cases tended to be chemosensitive (odds ratio=2.397) and less likely to be chemoresistant (odds ratio=0.417).
Similar analyses were performed for 310 TCGA patients who received platinum-based chemotherapy. There was a significant difference in sensitivity and resistance to platinum-based chemotherapy between the high- and low-risk groups (Fisher’s exact, p=6.59×10-4), as the number of chemosensitive and chemoresistant patients were 100 and 33 for high-risk groups, respectively, while in low-risk groups, the numbers were 157 and 20, respectively. Low-risk cases were more likely to be chemosensitive to platinum (odds ratio=2.591) and less inclined to develop chemoresistance to platinum (odds ratio=0.386) relative to the high-risk group.
Since only TCGA samples had drug response information, we used the GDSC database to predict the response to commonly used chemotherapeutic drugs for each training sample. As shown in Figure 3E, patients in the low-risk groups were more sensitive to Cisplatin, Gemcitabine, Oxaliplatin, Sorafenib, Tamoxifen, and Topotecan. In contrast, patients in the high-risk groups were more sensitive to 5-Fluorouracil, and there were significant differences between the high- and low-risk groups (p<0.05, Wilcox test).
The above results revealed a significant difference in tumor response to chemotherapy between the high- and low-risk groups predicted by SOV-P20."
And discussed the platinum resistance in the revised Discussion:
"Most serous ovarian cancer samples analyzed in this study were high-grade. Currently, the standard first-line care for high-grade serous ovarian cancer remains surgical resection and platinum-based chemotherapy. We have analyzed the tumor response to platinum-based chemotherapy. The results showed a significant difference in sensitivity to platinum-based chemotherapy between the high- and low-risk groups. Notably, resistance to platinum-based chemotherapy was present in both high- and low-risk groups, consistent with the assumption that a small proportion of ovarian cancer cells are chemoresistant from the beginning [42]. Such inherent resistance in ovarian cancer may occur due to reduced immunosurveillance and drug-resistant cells which belong to cancer stem cells (CSCs) [7, 42]. CSCs from the ovaries have a particular genetic signature that can engineer the original tumor mass, develop drug resistance, and promote recurrence. In addition to this possibility, platinum resistance may also emerge due to reduced intracellular drug accumulation, intracellular inactivation of the agent, increased DNA repair, or impaired apoptotic signaling pathways [43]. Therefore, the prediction of a tumor to platinum chemotherapy in ovarian cancer will be further research directions in the future."
Reviewer 2 Report
This Chinese study evaluated the role of SOV-P20 as a prognostic biomarker in serous ovarian cancer. I have the following comments:
-In the introduction section the authors should add information regarding the clinical prognostic factors for ovarian cancer.
-What kind of serous ovarian cancers have been studied? Low-grade or high-grade? Were there any borderline cases?
-Were age, grade and stage the only clinical factors included in the analysis? How many patients have been treated with chemotherapy and what was the response to treatment?
-The authors state that ‘clinical information was not available in all datasets, we only included the existing clinical elements in each dataset’. How many datasets had existing clinical information?
-Has SOV-P20 been prospectively validated? If not the authors should propose a protocol of a prospective study and provide a short description in the discussion section.
Reviewer 3 Report
I read with great interest the manuscript, which falls within the aim of this Journal. In my honest opinion, the topic is interesting enough to attract the readers’ attention. Nevertheless, the authors should clarify some points and improve the discussion, as suggested below.
Authors should consider the following recommendations:
- Manuscript should be further revised in order to correct some typos and improve style.
- One of the main important problems regarding ovarian cancers is the recurrence after surgery and first line chemotherapy. Usually, the recurrence of the disease poorly responds to first, and sometimes even second line chemotherapy. In this scenario, it is possible that the ovarian cancer inherent resistance may be due to reduced immunosurveillance and drug-resistant cells. Authors should discuss this hypothesis, referring to: PMID: 27995176; PMID: 26513872.
Author Response
Dear Editor and reviewer,
We are sincerely grateful for your editorial efforts and suggestions for our manuscript and the anonymous reviewer for his/her insightful comments on our manuscript, which are very helpful in strengthening this manuscript.
Following are our point-by-point responses to the reviewer’s comments, and the reviewer’s critiques have been accommodated fully in various parts of the revised manuscript (marked up using the “Track Changes” function in MS Word, in the reply shown in BLUE color).
We hope that you will now find the revised version of our paper suitable for publication in Diagnostics.
Thank you very much.
Sincerely yours,
Guini Hong
Reviewer 3
Comments and Suggestions for Authors
I read with great interest the manuscript, which falls within the aim of this Journal. In my honest opinion, the topic is interesting enough to attract the readers’ attention. Nevertheless, the authors should clarify some points and improve the discussion, as suggested below.
Authors should consider the following recommendations:
-Manuscript should be further revised in order to correct some typos and improve style.
Reply 1: Done as suggested. We have corrected all the typos and carefully revised the manuscript to improve the style.
- One of the main important problems regarding ovarian cancers is the recurrence after surgery and first line chemotherapy. Usually, the recurrence of the disease poorly responds to first, and sometimes even second line chemotherapy. In this scenario, it is possible that the ovarian cancer inherent resistance may be due to reduced immunosurveillance and drug-resistant cells. Authors should discuss this hypothesis, referring to: PMID: 27995176; PMID: 26513872.
Reply 2: Thank the reviewer for this critical comment. According to this suggestion, we have discussed this hypothesis in the revised Discussion: "Most serous ovarian cancer samples analyzed in this study were high-grade. Currently, the standard first-line care for high-grade serous ovarian cancer remains surgical resection and platinum-based chemotherapy. We have analyzed the tumor response to platinum-based chemotherapy. The results showed a significant difference in sensitivity to platinum-based chemotherapy between the high- and low-risk groups. Notably, resistance to platinum-based chemotherapy was present in both high- and low-risk groups, consistent with the assumption that a small proportion of ovarian cancer cells are chemoresistant from the beginning [42]. Such inherent resistance in ovarian cancer may occur due to reduced immunosurveillance and drug-resistant cells which belong to cancer stem cells (CSCs) [7, 42]. CSCs from the ovaries have a particular genetic signature that can engineer the original tumor mass, develop drug resistance, and promote recurrence. In addition to this possibility, platinum resistance may also emerge due to reduced intracellular drug accumulation, intracellular inactivation of the agent, increased DNA repair, or impaired apoptotic signaling pathways [43]. Therefore, the prediction of a tumor to platinum chemotherapy in ovarian cancer will be further research directions in the future."
Round 2
Reviewer 2 Report
All comments have been addressed and I
believe the manuscript is now suitable for
publication